# Social Behavioral Deficits in Krushinsky-Molodkina Rats, an Animal Model of Audiogenic Epilepsy

**DOI:** 10.3390/jpm12122062

**Published:** 2022-12-14

**Authors:** Anastasiya A. Rebik, Vyacheslav D. Riga, Kirill S. Smirnov, Olga V. Sysoeva, Inna S. Midzyanovskaya

**Affiliations:** 1Institute of Higher Nervous Activity and Neurophysiology, Russian Academy of Sciences, 117485 Moscow, Russia; 2Faculty of Biomedicine, Pirogov Russian National Research Medical University, 117997 Moscow, Russia; 3Vladimir Zelman Center for Neurobiology and Brain Rehabilitation, Skolkovo Institute of Science and Technology, 121205 Moscow, Russia; 4Center for Cognitive Sciences, Sirius University of Science and Technology, 354340 Sochi, Russia

**Keywords:** comorbidity of ASD and epilepsy, autistic-like behavior, Krushinsky-Molodkina rat strain, behavioral neurobiology

## Abstract

In clinical practice, epilepsy is often comorbid with the autism spectrum disorders (ASDs). This warrants a search of animal models to uncover putative overlapping neuronal mechanisms. The Krushinsky-Molodkina (KM) rat strain is one of the oldest inbred animal models for human convulsive epilepsies. We analyzed the behavioral response of adult seizure-naive KM males in three-chambered tests for social preference. We found that a presence of social stimuli (encaged unfamiliar Wistar rats of the same age and sex) evoked a reduced or reversed exploratory response in freely moving KM individuals. The epilepsy-prone rats demonstrated remarkably shortened bouts of social contacts and displayed less locomotion around the stranger rat-containing boxes, together with a pronounced freezing response. The decrease in social preference was not due to a general decrease in activity, since relative measures of activity, the index of sociability, were decreased, too. The susceptibility to audiogenic seizures was verified in the KM cohort but not seen in the control Wistar group. We propose the KM rat strain as a new animal model for comorbid ASD and epilepsy.

## 1. Introduction

The prevalence of epilepsy among the general population is approximately 0.6%, and in 2015, it was the fourth most severe neurological disorder worldwide. Similarly, in the early 2010s, the prevalence of ASDs among the general population was estimated at about 0.8% [1]. Epilepsy is often accompanied by autism spectrum disorders (ASDs), although the type of relation between these conditions is not well-established. According to a recent meta-analysis, about 10% of individuals with an ASD have epilepsy, and a similar percent of patients with epilepsy have an ASD; that is higher than in the general population [1]. Thus, shared genetic and neurophysiologic factors can link these conditions. The major diagnostic criteria for ASDs are social and communication deficits, as well as repetitive and stereotypical behavior (APA, [2]). At the same time, the constellations of these symptoms can be different and there are many other co-occurring symptoms, including sensory abnormalities, particularly hypersensitivity to auditory stimuli [3].

The study of rat lines genetically predisposed to epilepsy is promising for the emergence of new animal models with comorbidity of ASDs and epilepsy. Most of the main studies of the complex pathology of epilepsy and ASDs in laboratory rats were performed on pharmacological models of seizure provocations (for example, pilocarpine injections), which raises the question of the side effects of exposure to the chemical agent itself. It is preferable to use models with the hereditary nature of the disease, which makes it possible to investigate the mechanisms of pathology without externally introduced changes.

Phenotyping epileptic rat strains for autistic-like traits can shed light on the mechanism that underlies the relation between autism and epilepsy. For example, the FAST rat strain, which has an increased proneness to pharmacologically induced seizures, demonstrates various behavioral features associated with the phenotype of ASDs with hyperactivity [4,5]. At the same time, a number of other epileptic rat strains, such WAR and WAG/Rij rats, did not show social deficits in their behavior [6,7]. Here, we aim to phenotype one of the oldest rat models of audiogenic epilepsy, the Krushinsky-Molodkina (KM) strain [8], for social behavior which has not been rigorously studied in this regard. To segregate the genetic factors from those triggered by the seizure itself, we studied seizure-naive KM rats before they experienced their first seizure and only after the behavioral part of the experiment validated the proneness to audiogenic epilepsy.

## 2. Materials and Methods

### 2.1. Animals

The work was approved by the Institutional Ethical Committee and carried in accordance with the requirements of the ARRIVE guidelines.

Adult male rats of the KM strain (*n* = 12) and outbred Wistar strain (*n* = 16), aged 4–6 months and weighing 250–400 g were used in the study. The animals arrived at the animal chapter of IHNA at the age of 6–8 weeks. The cage contained 4-6 rats, with food and water ad libitum. All the animals were intact and drug-naive prior to the experiments.

### 2.2. Behavioral Tests

#### 2.2.1. Habituation

The behavioral experiments were preceded by 2 habituation sessions, when the animals were brought in their home cages to the experimental room for at least 2 h. The tests were scheduled to follow each other in 3–6 days, starting from the elevated plus maze (Section 2.2.2), followed by the 3-chambered social preference and social novelty tests (Section 2.2.4). The two social interaction tests were preceded by additional habituation sessions: the animals were allowed to investigate the experimental setup for 20 min in the absence of stimuli rats. After each session, the surfaces were cleaned and wiped with paper towels and 50% alcohol solutions.

The stimulus rats were habituated to being trapped in the stimulus box (see below, Section 2.2.2) for 30–60 min before the start of the social interaction experiments. The “empty” boxes were similar to the stimuli boxes in every respect, except that they contained the stranger animals.

#### 2.2.2. Elevated Plus Maze

A standard elevated plus maze test was done [9]. Briefly, the apparatus consisted of 2 open arms (10 cm× 46 cm), 2 closed arms (10 cm× 46 cm× 16 cm), and the central plate (10 cm× 10 cm). The scheme is shown in Figure 1. The maze was elevated for 90 cm from the floor surface. A video camera (Sony TRV75, Tokyo, Japan) was attached at about 120 cm above the maze. The experiment was recorded by a computer in an adjacent room.

The rats were placed individually on the central plate, headed to an open arm and with its back to the experimenter; the landing was a signal to start the tracking. The animals were allowed to investigate the maze for 10 min.

#### 2.2.3. Three-Chambered Social Preference/Novelty Tests

Next, the 3-chambered test for social preference and social novelty were used (reviewed in [10,11]). The apparatus consisted of a 60 cm× 60 cm black arena, with black 60 cm high wooden walls and dividers, sectioning the box into two chambers and a rectangle compartment between them (Figure 1b). Each divider had an opening, allowing free passages between the sections. Two dark plastic perforated boxes (as described above, see Section 2.2.2) were placed into each chamber. First, the setup included an empty box and a stranger rat-containing box. The animals were allowed to investigate the environment for 10 min.

Next, the empty box was changed for an identical container with a new rat, to test the response to a social novelty (Figure 1c). The next 10 min of behavior were registered, as described before. Below, the sessions are referred as the “social preference” and “social novelty” tests.

#### 2.2.4. Off-Line Analysis

All sessions were videorecorded with a digital camera linked to a computer in an adjacent room. The recordings were digitized and analyzed offline automatically [track lengths, freezings, times spent in virtually defined marked areas] and offline by blind experts [contact time, rearings, grooming episodes].

A free software ToxTrac (version 2.95, Umea, Sweden) was used for offline tracking the moving animals in all tests. Tracking was done using ToxTrac’s ID algorithm Toxld (Ssi.Rep.). The algorithm used allowed us to track the rostral part of the body of the freely moving animal (which was important when evaluating social contacts). The following measurements were automatically taken: time spent in defined zones, total distance travelled in the defined zones, number and time of freezing episodes [12].

The sociability index was calculated as the ratio:$$\frac{The\;rat\;box\;exploration\;time}{The\;empty\;box\;exploration\;time}$$

#### 2.2.5. Verification of Audiogenic Seizure Susceptibility (AGS)

Upon completion of the behavioral testing, audiogenic seizures were provoked as described before [13,14]. As expected, in all KM rats, the sound provocation evoked full-blown seizure fits, which quickly evolved into clonic and tonic phases (rev. in [8,15]). All Wistar individuals remained seizure-free.

#### 2.2.6. Statistical Analysis

The main method for assessing the significance of differences between the experimental and control groups was the Mann–Whitney U-test; within group effects were estimated by the Wilcoxon test.

## 3. Results

### 3.1. Elevated Plus Maze

KM rats displayed a decrease in motor activity and increased anxiety. Both horizontal and vertical elements of the exploratory behavior were reduced, and the time spent in the open arms was shorter compared to the control group (Figure 2a–c). The total number of visits to all arms decreased (Figure 2g). The elevated anxiety signs were seen as more frequent episodes of facial grooming and freezing (Figure 2d,e; Appendix A).

### 3.2. Three-Chambered Social Preference/Social Novelty Tests

The KM group demonstrated hypolocomotion and reduced time spent in the stimuli-enriched chamber, together with longer visits to the empty chamber (Figure 3a,b). duration (Figure 4a) and number of contacts (Figure 5a) were dramatically low in the KM group. Freezing episodes were prolonged significantly compared to Wistar group (Figure 4d). The sociability index (T stim/T empty) was significantly lower (*p* < 0.01) in the group of KM rats (0.51 ± 0.12 vs. 1.38 ± 0.27, *p* = 0.01; see also Appendix A).

In the next session, the so-called social novelty test, the KM males preferred the same location that they had chosen for the first session (green brackets in Figure 3c). The time spent in each zone did not differ between the tests in the KM groups (Wilcoxon matched pairs test, *p* > 0.05). In contrast to this, the Wistar rats increased the time spent in the new “habituated” zone (*p* < 0.01, Wilcoxson matched pairs test) compared to its previous “empty” state (see also Appendix A).

The number of social contacts demonstrated by KM group was very low again (Figure 5a). Investigatory activity (judged by the number of rearings) was reduced in the KM rats compared to the control group. Remarkably, in the KM rats, the freezing response dominated and occupied almost all the time of the “social novelty” session (Figure 4d, see also Appendix A).

### 3.3. AGS Provocation

After the completion of the behavioral testing, AGS proneness was checked in the rat groups. The sound provocation evoked full-blown seizure fits in all the KM rats, which agrees with the main breeding criterion for the strain (rev. in [8,15]). Namely, a wild running started with a latency of 10–12 s, which quickly evolved into clonic seizures and, in 30–45 s, ended up with the tonic phase, with all limbs extended. In Wistar rats, the same sound administration did not induce paroxysmal fits, but evoked a regular behavioral response (from short freezing to walking, rearings, and sniffing), in line with that reported earlier [13,14].

## 4. Discussion

The results presented strongly support a hypothesis on consistent social deficits in KM rats. Time spent in social interaction, as well as the number of approaches to it, were remarkably reduced in the KM group (Figure 4a and Figure 5a). Somewhat paradoxical results were seen in the “social novelty” paradigm: a newly added conspecific provoked a weak increase in a number of contacts in the KM group (Figure 5a). However, the number and duration of contacts were still dramatically low. Interestingly, that KM rats showed the same distribution of zonal times, i.e., visits to the chambers, did not change significantly on addition of a “new” conspecific (see Section 3.2). The control rats reacted to the added conspecific by a significant increase in the corresponding zonal time (Figure 3b, Section 3.2). Two social stimuli presented in the “social novelty” session provoked an enhanced freezing response in the KM rats (Figure 4b), with the duration of freezing comparable to the total time of the “social novelty” session (Section 3.2).

We confirmed several previously discovered behavioral features of KM rats, such as increased anxiety and reduced locomotor activity [7,16], and described the social deficits in these rats for the first time. The existing rat models with autistic-like traits mainly demonstrate increased locomotor activity and impulsivity (valproate rats, FAST kindling rats), which attribute them to the phenotypes of comorbid ASD and ADHD in humans [17,18,19,20,21]. At the same time, the hypoactivity symptom of ASD has also received more attention in recent years, as it is reported in about 40% of individuals with an ASD [22]. The KM animal in our study shows evident signs of hypoactivity, together with social contact avoidance, making them more similar to some murine models of monogenic ASD-associated syndromes, which also showed signs of home-cage hypoactivity [23].

Given the heterogeneity of idiopathic ASDs, the determination of the mechanisms leading to ASD–epilepsy comorbidity can help in the stratification of autism. Since the causes leading to the idiopathic ASDs are poorly understood, progress on this path is largely based on the study of genetic syndromes associated with autistic traits (reviewed in [24,25]), but definitely requires validated animal models reproducing certain clinical phenotypes.

## 5. Conclusions

Here, we show the social deficits in inbred KM rats, genetically susceptible to audiogenic seizures. Thus, this rat strain can be considered as a potential animal model of ASD–epilepsy comorbidity, these rats also showed hypoactivity. Future experiments are planned for more extensive phenotyping of this model and the exploration of the putative effects of seizure experience on the behavioral traits of KM animals.

## Figures and Tables

**Figure 1 jpm-12-02062-f001:**
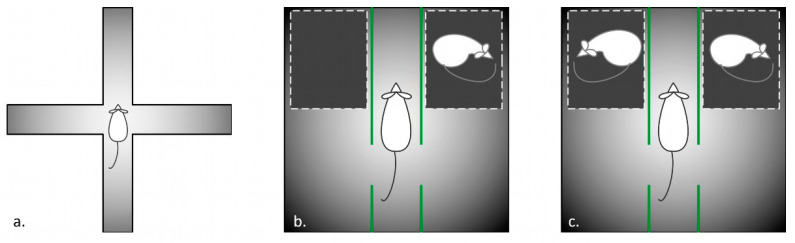
Schemes of experimental setup: (**a**) elevated plus maze; (**b**) three-chambered social preference test; (**c**) three-chambered social novelty test.

**Figure 2 jpm-12-02062-f002:**
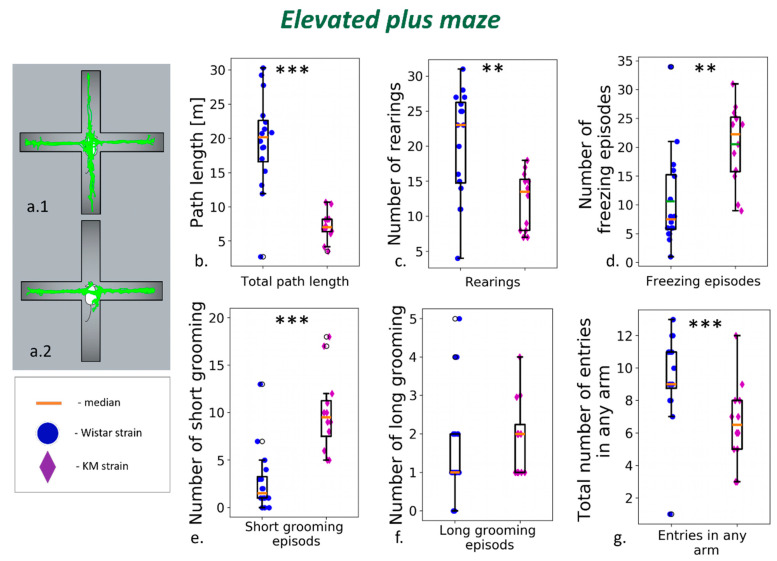
Behavior of KM and Wistar rats in the elevated plus maze test: (**a.1**) Examples of individual tracks of Wistar rat and (**a.2**) of KM rat. (**b**) The total path length and (**c**) rearings were decreased in KM (violet diamonds) compared to Wistar (blue circles). (**d**) Episodes of freezing and (**e**) number of facial grooming episodes were on the contrary increased in KM compared to Wistar. (**f**) Whole-body (long) grooming was similar among groups. (**g**) The total number of entries to any arm were also attenuated in KM. Data are represented as the box plot and whiskers (indicating variability outside the upper and lower quartiles). Statistical significance is indicated as follows: *** *p* < 0.001, ** *p* < 0.01.

**Figure 3 jpm-12-02062-f003:**
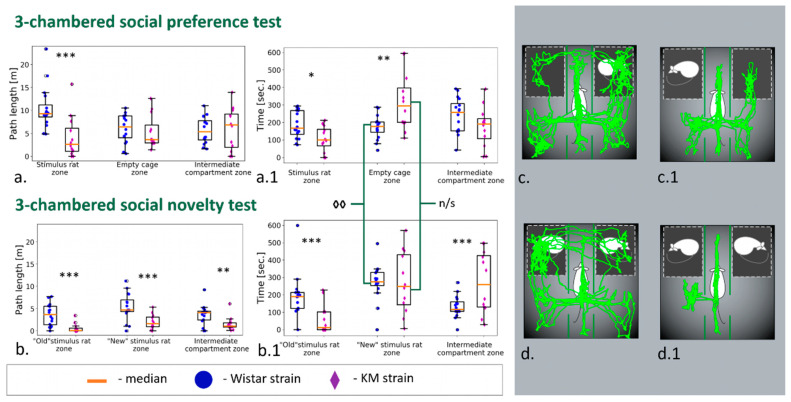
Locomotion activity in social tests: path lengths (**a**) and the zone times (**a.1**) in three-chambered social preference test; in social novelty test (**b**,**b.1**); representative individual tracks of Wistar (**left**) and KM (**right**) rats in the three-chambered social preference test (**c**,**c.1**), in the three-chambered social novelty test (**d**,**d.1**). Wistar rats are shown as blue circles, KM as violet diamonds. KM rats demonstrated a significant decrease in the path travelled around the stimulus rat zone (**a**), together with the corresponding time spent (**b**). The addition of the second conspecific (the three-chambered social novelty test) induced general hypolocomotion in KM rats (**b**). Remarkably, after the addition of the new social stimulus, Wistar (but not KM) rats significantly increased the corresponding chamber time (green brackets mark the within-group comparisons, (**a.1**,**b.1**)). Data are represented as the box plot and whiskers (indicating variability outside the upper and lower quartiles). Between-group difference (Mann–Whitney U-test) is indicated as follows: *** *p* < 0.001, ** *p* < 0.01, * *p* < 0.05. Within-group difference (Wilcoxon matched pairs test) is indicated as ◊◊ for *p* < 0.01.

**Figure 4 jpm-12-02062-f004:**
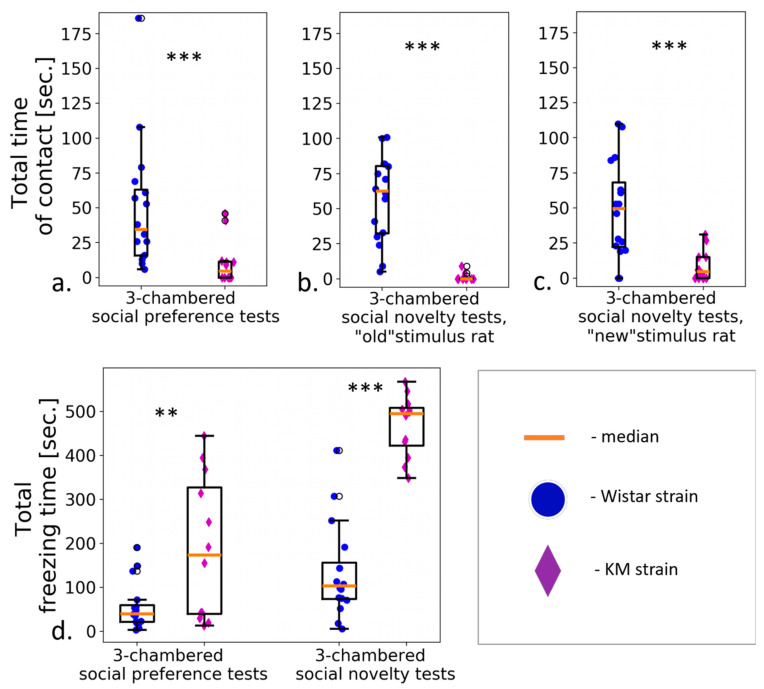
All types of social contacts were shorter in KM rats (violet diamonds) compared to Wistar rats (blue circles), in contrast to the freezing response: (**a**–**c**) total contact time with unfamiliar rats; (**d**) total durations of freezing. Data are represented as the box plot and whiskers (indicating variability outside the upper and lower quartiles). Statistical significance is indicated as follows: *** *p* < 0.001, ** *p* < 0.01.

**Figure 5 jpm-12-02062-f005:**
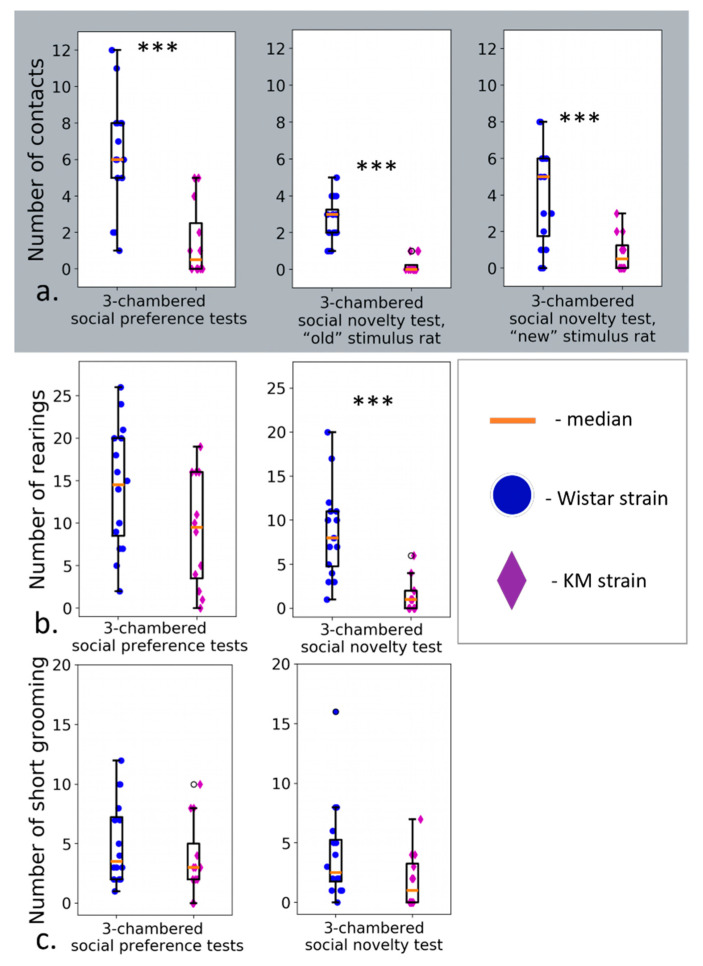
Numbers of all types of contacts were lower in KM group compared to Wistars (**a**). Rearings (**b**) were reduced, and short grooming bouts (**c**) did not change (**c**) in KM rats (violet diamonds) compared to Wistar rats (blue circles) in the social preference/social novelty tests. Data are represented as the box plot and whiskers (indicating variability outside the upper and lower quartiles). Statistical significance is indicated as follows: *** *p* < 0.001.

## Data Availability

The experimental data are fully reported in the manuscript and Appendix A. Additional information is available upon a reasonable request.

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
