# Peer review of "Social Behavioral Deficits in Krushinsky-Molodkina Rats, an Animal Model of Audiogenic Epilepsy"

_jpm, 2022, doi:10.3390/jpm12122062_

Round 1
Reviewer 1 Report
Thank you for inviting me to review this manuscript.
The incidence of Audiogenic epilepsy in human studies is actually low. However, the author aimed to find an animal model representing epilepsy and ASM. This finding is important to many researchers that want to study epilepsy and ASM using animal models.
The author should clarify in the introduction why they used audiogenic epilepsy animals rather than the kainate-induced seizures animal models.
Author Response
Thank you for your review.
We added a paragraph about the prospects and advantages of finding new genetic models of comorbidity of ASD and epilepsy.
Reviewer 2 Report
The data seems convincing that there is a difference in social interactions between KM and Wistar rats. The major shortcoming of the paper is that the KM mouse does not appear to be genetically linked to autism (e.g. there are no given mutations in any genes that have been associated in human autism). I’m not sure how or if this could be corrected as autism is a syndrome and the genetic causes of it are still not well understood. The authors state that this could be used for studying comorbidities (autism and epilepsy) and based on the phenotype, perhaps they are correct. More details about how the observed social behaviors in KM rats mimic those in seen autism would help.
I have two specific comments:
In the first paragraph: What is the prevalence of ASD and epilepsy in the general population? The statement could be strengthen if the prevalence of ASD and epilepsy in the general population were given.
Line 198: “…did not changed…” should be “…did not change…”
Author Response
Thank you for your review.
We will consider in more detail the behavioral features of KM stain rats in future articles.
We corrected the grammatical inaccuracy.
And we also added current information about the prevalence of epilepsy and ASD.